# A Digital Math Game and Multiple-Try Use with Primary Students: A Sex Analysis on Motivation and Learning

**DOI:** 10.3390/bs14060488

**Published:** 2024-06-08

**Authors:** Claudio Cubillos, Silvana Roncagliolo, Daniel Cabrera-Paniagua, Rosa Maria Vicari

**Affiliations:** 1Escuela de Ingeniería Informática, Pontificia Universidad Católica de Valparaíso, Valparaíso 2362807, Chile; silvana.roncagliolo@pucv.cl; 2Escuela de Ingeniería Informática, Universidad de Valparaíso, Valparaíso 2362905, Chile; daniel.cabrera@uv.cl; 3Instituto de Informática, Universidade Federal do Rio Grande do Sul, Porto Alegre 90010-150, Brazil; rosa@inf.ufrgs.br

**Keywords:** computer-based learning, sex differences, multiple-try feedback, mathematical training

## Abstract

Sex differences have been a rarely addressed aspect in digital game-based learning (DGBL). Likewise, mixed results have been presented regarding the effects according to sex and the conditions that generate these effects. The present work studied the effects of a drill-and-practice mathematical game on primary students. The study focused on an analysis by sex, measuring motivation and learning in the practice activity. Also, two instructional mechanics were considered regarding the question answering to search for possible differences: a multiple-try feedback (MTF) condition and a single-try feedback (STF) condition. A total of 81 students from four courses and two schools participated in the intervention. The study’s main findings were as follows: (a) the girls outperformed the boys in terms of the students’ learning gains; (b) the girls presented lower levels of competence and autonomy than the boys; (c) under MTF, the girls presented lower levels of autonomy but no differences in competence contrasted with the boys; (d) under STF, the girls presented lower levels of competence but no differences in autonomy contrasted with the boys; (e) no sex differences existed in interest, effort, and value, in general, as per the instructional condition. This study enhances the knowledge of sex differences under diverse instructional settings, in particular providing insights into the possible differences by sex when varying the number of attempts provided to students.

## 1. Introduction

The rapid advancement of technology, particularly in computing and information technology (IT), has significantly impacted education, with various tools now available to support learning processes, ranging from Learning Management Systems (LMSs) like Chamilo, Moodle, and H5P and educational game generators (e.g., Educandy, Educaplay) to specific educational math platforms, like photoMath and Symbolab. These tools have been shown to significantly influence learner motivation, attitudes, and achievement, especially in mathematics [1]. However, sex differences in technology use and learning have been observed, with males showing more interest in using computers and technologies than females [2,3]. Males also perceive multimedia technology for learning as more beneficial than females [3].

In terms of cognition and learning, studies suggest that females perform better than males in planning and attention tasks [4,5]. However, the research also indicates that men and women employ distinct strategies in planning tasks [5,6], with women having higher verbal and memory skills, while men exhibit stronger capabilities in spatial cognition and learning, and both develop different approaches to complex planning tasks throughout their lives. This suggests that sex is a key factor in the efficient and effective use of educational digital technologies.

Integrating gaming with learning is not an easy task, as even setting up any of the above-mentioned learning platforms is not straightforward. Many of these tools allow for the configuration of different instructional features, such as question types, number of attempts, and types of feedback. However, a key problem in their use is that it is not always clear which combinations of all these available characteristics are most suitable for a specific group of students. Adding the factor of sex differences makes it even less clear which of these characteristics generate sex-neutral effects and which do not. While studies on digital game-based learning (DGBL) have extensively researched various emotional and motivational factors, studies covering sex differences in learning gains and engagement [4,7,8], attitudes [9,10], motivation [11], and anxiety [12,13] have shown contradictory findings. 

This study aimed to address these gaps by exploring the motivational and learning effects of using a digital math game on primary students. It focused on identifying possible differences by sex, especially when varying the number of attempts provided as part of the instructional design. This research will contribute to a deeper understanding of how sex influences learning outcomes in DGBL, ultimately aiding in the development of more effective and inclusive educational technologies.

The next section provides a background on the existing literature covering sex differences in digital learning contexts and the inclusion of motivation and the related affective outcomes for deepening the concepts of instructional feedback and multiple attempts. Then, a sub-section on the Self-Determination Theory is presented to frame the motivational constructs used in our study. Section 3 presents the objectives and research questions guiding the current study together with a detailed description of the experiment’s materials and methods, including the MatematicaST game, participants, instruments, and procedure used. Then, Section 4 and Section 5 show the results and discussion, and Section 6 presents the conclusion with the main findings and their implications.

## 2. Background

### 2.1. Sex Differences and Learning in Digital Contexts

The exploration of sex differences in learning reveals that males and females interact with technology in distinct ways. Teen girls, for instance, tend to spend more time on smartphones and engage more with social networks, while teen boys are more inclined to spend their time playing games on electronic devices [14]. This divergence in technological engagement extends to mobile learning settings, where males appear to be more influenced by social factors than their female counterparts [15]. This study focuses on the biological sex of students rather than their sexual identification or orientation. Despite the above, students had to write their gender identification on the perception questionnaire, not registering any gender–sex difference. Therefore, it seems that all participants did identify with their sex.

Delving deeper into the field of gaming, a sex-based dichotomy becomes evident. Males often perceive games as unique and engaging experiences, which they find relaxing and conducive to better performance. This association of diversion and catharsis with gaming is typically more pronounced in males [16]. In contrast, it seems that females view games differently. Rather than seeing them as memorable or entertaining experiences, females tend to regard games as just an alternative learning method [16]. Furthermore, females are frequently more skeptical than males about the instructional potential of games, indicating a more cautious approach to the integration of gaming into learning [16].

When focusing on academic performance, research indicates that females generally outperform males, exhibiting superior academic achievements [17] and excelling in learning assignments [18]. This trend extends to students’ disposition towards risk, with a study by Cipriani [19] revealing that females tend to be more cautious in multiple-choice tests, omitting more items and guessing less than males. This behavior is attributed to males being more risk-prone, interpreting risk as a challenge rather than a threat.

Despite these observed differences, there is a scarcity of studies investigating sex differences in gameplay and digital game-based learning (DGBL) [8]. However, the existing body of research suggests that females may have an edge over males in terms of engagement and learning gains in DGBL [8,20,21]. For instance, Lukosch et al. [4] found that female participants outperformed males in a game designed to develop planning operation skills in a container terminal setting.

Similarly, a study by Yeo et al. [22] on fifth-grade students in Taiwan revealed that females showed greater improvements in learning performance than males when using a digital food-chain game. Similar results were also observed in a study by Khan et al. [8], which involved an eighth-grade chemistry (reactivity) game-based learning application. The study found that the girls outperformed the boys in the post-test scores under the educational application, while no gender differences were observed under traditional science instruction. These findings lend further support to the notion of sex-based learning differences.

However, it is worth noting that other research [10,11,23] has reported no significant learning differences by sex. For example, a study by Chung and Chang [11] involving a digital game for first-aid training in English for Chinese non-native speakers found no significant sex differences in the learning achievements. These contrasting findings underscore the need for further investigation to clarify under which conditions one sex may have a learning advantage over the other. The collective insights from these studies justify and highlight the need for further research in this area to better understand and cater to these distinct learning preferences.

### 2.2. Sex, Motivation, and Related Affective Outcomes

The role of sex in education, particularly in the context of motivation and affective outcomes, has been the subject of extensive research. However, past studies have revealed the complex interplay of sex, learning performance, and motivation in education. In the context of student attitudes towards e-assessments, Bahar and Asil [9] found that males reported more positive attitudes than females. However, when it comes to students’ interest in educational video games, Manero et al. [24] found no sex differences in students’ interest in theater-going. The researchers explored the impact of sex, age, and gaming habits on the effectiveness of the game “*La Dama Boba*” (*The Foolish Lady*), a graphical adventure based on the theater play of the same name. Interestingly, in the study of the digital game for first-aid training (EFA) by Chung and Chang [11], girls presented higher motivation levels than boys, although they had similar learning levels. The authors argue that their game was gender-moderated (or balanced), as the three learning activities used elements such as storylines, challenges that did not emphasize competition, educational values, fun factors, and interactivity that offered game exploration opportunities, enhancing female motivation. They also measured usability with a System Usability Scale (SUS) test, finding a non-significant higher score for females.

A study by Khan et al. [8] measured students’ engagement with four factors: positive body language, consistent focus, confidence, and fun and excitement. They found that girls in the learning application presented higher engagement levels than boys, although this was not significant, by using the “student engagement walk-through checklist” [25] administered by teachers during the intervention. In the experience with the food-chain digital game of Yeo et al. [22] with primary students, they determined that the attention of the medium- and high-prior-knowledge (PK) groups was significantly higher than that of the low-PK group. Also, the medium-PK group reported higher satisfaction levels than the low- and high-PK groups. The results also varied by sex. In the high-PK group, males presented higher attention levels than females, while in the low-PK group, males showed higher confidence than females. The authors used the Attention, Relevance, Confidence, and Satisfaction (ARCS) Learning Motivation Scale [26], which is based on the expectation value theory and individuals’ success expectations.

A recently explored variable has been student anxiety, especially in math [27,28] and digital game-based learning [29]. Diverse authors [12,30] found no sex differences concerning anxiety and achievement in math. The Goetz et al. [31] study examined possible sex differences between trait (habitual) anxiety and state (transitory) anxiety in mathematics in nearly 700 students. They concluded that females presented a higher trait of math anxiety compared to males, but no differences existed for state anxiety using experience-sampling methods in activities like taking a math test or attending math lessons. Another relevant finding was that the students’ self-perceptions of their competence in mathematics were lower in girls than boys despite them having similar grades in math. The authors argue that this difference in perceived competence helps to partially explain sex differences among state-trait anxiety. More recently, Wang [13] suggested that the student’s spatial ability can be a relevant factor in mediating math anxiety when considering sex differences.

In conclusion, the effects of sex on learning and motivation are not straightforward. Different variables and conditions make it hard to draw consistent conclusions from the revisited literature. Diverse authors [8,10,20] highlight the need for more research focusing on sex, learning approaches, and motivation for building strategies that foster sex parity in DGBL and education.

### 2.3. Multiple-Attempt Use with Instructional Feedback

Instructional feedback can be defined as providing students with information to correct their answers, aiding in error identification, misconception correction, and the enhancement of problem-solving strategies and self-regulation [32,33,34]. In a broader sense, feedback encompasses any information provided by an agent—such as a teacher, peer, book, parent, or experience—regarding a person’s performance or understanding [35]. In this study, the instructional feedback was centered on the task level and its formative dimension (learning), diverging from delayed summary feedback [36] and feedback focused on task motivation or self-regulation processes [34].

Feedback provision can vary widely [37], with three feedback types being most frequently used in game-based learning settings during the last decade [34,38,39] as part of single-attempt instructional feedback (STF) implementations. Knowledge of Result (KR) confirms the correctness of students’ answers or marks errors without additional information [34]. Knowledge of Correct Response (KCR) specifies the correct answer without further explanation [40]. Elaborated feedback (EF) provides information on why an answer is right or wrong, offering explanations, additional materials, hints, or a combination of these [39,40]. Additionally, feedback types differ based on the number of attempts allowed. Answer-Until-Correct (AUC) feedback permits multiple attempts until the correct answer is reached, offering KR feedback between attempts [41]. Multiple-try feedback (MTF) allows a limited number of attempts, typically providing KR but sometimes offering KCR after the final try or hints on the first attempt [42].

Past literature has progressed in leveraging under what conditions each type of feedback seems better but has mainly focused on single-try (STF) alternatives. A study by Van der Kleij, Feskens, and Eggen [33] revealed that much research exists on elaborated feedback combined with KR or KCR in computer-based environments. Their meta-analysis showed that different feedback types were moderated by the kind of learning to be achieved, with EF outperforming both KCR and KR under high-order learning outcomes (LOs), while no major differences existed among these in low-order learning that involved verbatim or recall of information.

The above highlights the relevance of task complexity. By considering that feedback’s primary importance is the correction of errors, one would expect to see larger effects for instruction with higher error rates [43]; that is, more difficult topics have more possibilities for learning improvement. Differences in single-try feedback alternatives (KR, KCR, and EF) appear only upon student error.

Now, regarding multiple-attempt use, there is much less literature, and it is outdated, in addition to providing mixed results on the learning effects. Some studies show significant learning gains with multiple-try solutions [36,42,44], while others find no significant differences when compared to single-try alternatives [45,46,47]. Only the review by Clariana and Koul [44] compared MTF to different feedback types (KR, KCR, and EF), determining that MTF outperformed the other feedback types for higher-order learning outcomes (effect size—ES: 0.11) while being equivalent to or inferior for lower-order outcomes (ES: 0.22). It seems that in situations in which the test items involve comprehension and understanding rather than simply recall, such as in mathematical problem solving, the invitation to try again offers a chance for the elaboration and reorganization of information, potentially enhancing learning [42].

Past research has provided different arguments and used diverse theories to explain multiple-try effects. On the one hand, some research shows that the use of multiple trials (AUC and MTF) offers an interactive mechanism [38] for students’ errors, giving them multiple immediate exposures to the same item [48]. From a contiguity theory perspective, multiple trials seem beneficial compared to single attempts because they may engage learners in additional active processing following errors [49,50]. It is also in line with the information-processing theory, which argues that the continued engagement with a question needed upon an incorrect response can offer potential advantages [51,52], and with the idea that providing the correct answer after only one response, as happens with KCR, may “short circuit” learning [49,53].

On the other hand, some research argues that repeatedly asking a learner to answer the same question can be frustrating [49,54]. Providing multiple attempts at errors might encourage deeper thinking about the lesson unless the learner falls on random guessing because of frustration or impatience [55]. Furthermore, diverse past studies support the idea that learners, particularly those with low academic performances, can feel frustrated when lessons employ multiple-try feedback [55,56,57].

Referring to Salomon and Globerson’s concept of mindfulness [58], feedback can aid learning when received mindfully but might hinder it if it promotes mindlessness [42]. Along this line, besides knowing that MTF seems to be more beneficial for high-level learning outcomes, there is not much clarity regarding what other factors and their interactions promote mindful or mindless trial-and-error behavior when using multiple attempts.

### 2.4. Motivation and Self-Determination

The Self-Determination Theory (SDT) [59,60] is a psychological framework for explaining the motivation construct that is frequently used in DGBL. However, its use for explaining the effects of multiple trials has been limited. To the best of our knowledge, this is the first effort to apply the SDT to explain sex differences when using multiple attempts under DGBL. The SDT coins the concept of intrinsic motivation, the innate psychological need for competence, autonomy, and relatedness [61]. Competence involves the need for effectiveness in interactions with the environment [61], while autonomy regards the experience of free choice and “freedom in initiating one’s behavior” [62]. Relatedness refers to the fundamental human need for belongingness and connection with others [62]. In addition, the Cognitive Evaluation Theory (CET) (an SDT sub-theory) explores factors that either enhance or hinder intrinsic motivation [60]. The CET states that fostering intrinsic motivation in an educational setting involves providing diverse stimulating instructional elements and adequate challenges and promoting learner initiative and autonomy without control or pressure [59]. In this sense, activities with optimal challenge and effort levels enhance motivation [61]. Adequate challenges lead to effort, generating feelings of competence, while the absence of controlling conditions can positively impact effort levels [59]. Therefore, the SDT-CET make use of the following constructs to assess motivation: interest, competence, pressure, effort, choice, and value. All of them are positively correlated to motivation except for pressure, which is the reason why it is often used in reverse mode as no pressure.

Research on game-based learning has used the SDT with diverse focuses. Liao, Chen, and Shih [63] studied the relationship between cognitive load, motivation, and learning outcomes, while the authors of [64] studied the properties of scaffolds on intrinsic motivation. The SDT was also used to explain the relation of gamification techniques with motivation in virtual reality [65], and for bridging learning mechanics and game mechanics in GBL contexts [66]. Liu, Wang, and Lee [67] found that game quality and feedback significantly influenced the motivation and learning performances of students.

In our previous study utilizing MatematicaST [68], the focus was on contrasting multiple-try feedback and the single-try condition with KCR among primary-school students. The results revealed that MTF provided higher learning and higher levels of perceived competence and autonomy than the single-try condition with KCR. Also, multiple-try feedback presented an increase in perceived pressure. No significant differences were observed in terms of the perceived effort and value between the conditions, and both remained consistently high. As well, no sex analysis was conducted, as it was not a factor or focus of the paper.

In summary, the SDT has been applied to gamification, game mechanics, and game quality in computer-based learning contexts, but without controlling the instructional feedback, such as single-try and multiple attempts. There is little empirical research examining the SDT constructs with the sex and feedback types under DGBL contexts, showing the necessity of advancing the literature on how sex affects students’ motivation and the related affective dimensions, such as effort, pressure, and value, and also on how the feedback type and multiple-attempt inclusion could affect sex outcomes in terms of learning and motivation.

## 3. Materials and Methods

### 3.1. Study Objectives and Research Questions

Based on the above, the present study first aimed to explore the impacts of sex when using a digital math game with primary students. Second, it intended to assess whether such impacts differ across the two instructional feedback conditions (namely, MTF and STF). The study focused not only on the performance effects (learning gains) but also on the motivational effects based on the SDT-CET and the operationalization in terms of the interest, competence, autonomy, effort, no-pressure, and value constructs. Three high-order math-learning objectives were selected in this drill-and-practice training game. Therefore, based on the research gaps leveraged from the past literature, the following research questions were elaborated to guide the present investigation:
RQ1: What are the differences in the learning outcomes between male and female students using MatematicaST?RQ2: What differences exist in motivation in terms of interest (RQ2a), competence (RQ2b), and autonomy (RQ2c) between male and female students using MatematicaST?RQ3: What are the differences in the perceptions of effort (RQ3a), no pressure (RQ3b), and value (RQ3c) between male and female students using MatematicaST?RQ4: What differences exist in the effects of sex between the two feedback conditions (MTF and STF)?


### 3.2. MatematicaST Game

“MatematicaST” is a web platform with mathematical games, developed by our research group starting in 2017 as part of a Research & Development project. It has been piloted in elementary school courses in Chile, ranging from the 3rd to the 6th grade (from 8 to 11 years old), across various schools to practice mathematical topics [68]. For the current intervention, three math mini-games were offered: number identification, number ordering, and money counting (see Figure 1), aligned with the national curriculum for the 3rd and 4th grades in Chile. The number identification game exercises the place value on the symbolic representation of numbers with units, tens, and hundreds. The number-ordering game focuses on arranging four numbers from smallest to largest up to 3 digits. Players can drag the numbers to rearrange them and confirm the obtained sequence. The money-counting game focuses on counting coins of different amounts considering coins with values ranging from 1 to 500 (in CLP).

In each game, the question appears at the top and can be listened to by pressing the play button. Players have three lives (represented by red hearts) and earn ten points for each correct answer. Total points are displayed at the upper right. The platform also features a list of high scores on the right side.

The games have 2 versions, which vary the type of instructional feedback provided to students. In the STF version of the games, students have one opportunity to answer each exercise. After their response, they receive feedback indicating whether the answer is correct or incorrect, along with the correct solution in case of a wrong answer. In the MTF version, students have three attempts per exercise without losing a life. After each attempt, immediate feedback on their answers’ correctness (KR) is given. The correct answer is not revealed even after the last try. For more details on MatematicaST games and versions, please refer to [68].

### 3.3. Participants and Design

In this study, students from the third and fourth grades from three schools located in the Valparaíso region of Chile were engaged in the intervention, as the learning outcomes covered by the games corresponded to the mathematical topics covered in these grades, as part of the national curriculum. Initially, the study considered the participation of 95 students. However, after controlling for complete participation (student absences on activity days, missing written parental consent, or incomplete responses in post-test or post-IMI), the final number of subjects was 81.

The research followed a randomized pre-test–intervention–post-test design, with the sex (male/female) being the first factor of analysis and the number of attempts with the feedback type provided as a second factor (STF/MTF). The STF condition considered a single try with Knowledge of the Correct Response after answering. The MTF condition involved multiple tries (3 attempts) with Knowledge of the Result among the trials. Students were randomly assigned to these conditions, with 41 participants (23 male, 18 female) in the multiple-try group and 40 participants (19 male, 21 female) in the STF condition group.

The experiment involved private and subsidized schools, with high and medium socioeconomic statuses (SESs), respectively. However, students from public schools (low SESs) were missing and so should be considered in the next studies. In Chile in 2022, 56% of students went to subsidized schools, 35% went to public schools, and the remaining 9% went to private ones [69].

### 3.4. Procedure

The study was conducted during two subsequent school days on math lessons. On day 1, students were taken to the computer lab or provided with notebooks depending on the school facilities. They were initially given a motivational talk outlining the experiment’s objectives and clear instructions about the procedures and the tests to be administered (10 min). Subsequently, students took the pre-IMI-test and the learning pre-test using a paper-and-pencil format for 20 min. Following this, students accessed the web platform and engaged in the MatematicaST game activity on school computers for 1 h. On day 2, students responded to the post-IMI-test and the learning post-test (20 min).

### 3.5. Instruments

#### 3.5.1. Math Tests

The pre- and post-tests in this study were designed based on three specific learning objectives outlined in the Chilean mathematics program and its corresponding guide textbook. Collaborative efforts were made with teachers from participating schools, who actively contributed to the construction of these tests. Their valuable feedback was utilized to improve the instructions, exercises, and testing conditions. Each pre-test and post-test encompassed 18 items and six exercises related to each learning objective, which included tasks focused on (a) number identification, (b) number ordering, and (c) money counting. The tests were structured to have a maximum of 9.0 points, ensuring a comprehensive assessment aligned with the targeted learning outcomes.

#### 3.5.2. Motivational Questionnaire

The measurement of motivation following the Self-Determination Theory and its Cognitive Evaluation Theory was operationalized using the Intrinsic Motivation Inventory (IMI) [70]. The IMI assesses various constructs related to intrinsic motivation during specific activities. It should also be noted that its reliability and validity in measuring these constructs have been proven through numerous studies focusing on intrinsic motivation [7,64,66,68]. This research incorporated six key motivation constructs related to students’ self-perceptions: interest, competence, pressure, effort, choice (autonomy), and value. The perception tests consisted of 16 sentences rated on a 5-point Likert scale [71], ranging from 1 (“strongly disagree”) to 5 (“strongly agree”).

## 4. Results

The data were processed with SPSS software, version 29, for the statistical analyses and graph generation. Partial eta squared (*η*^2^) was used as a measure of the effect size, with values of 0.01, 0.06, and 0.14 for small, medium, and large effect sizes, respectively, provided by Cohen [72]. First, Table 1 shows the descriptive statistics for the pre- and post-test scores per sex and condition. Then, to assess the overall learnings and the possible effects of sex, a two-way repeated-measures analysis of variance (ANOVA) was performed considering sex as a between-subject factor and the (pre/post) test type as a within-subject factor.

### 4.1. Learnings by Sex

The results show a difference in the pre and post-test scores with F(1, 79) = 19.380, *p* < 0.001, *η*^2^ = 0.197, meaning that there were significant learning gains with the activity. The main effects of sex on learning resulted in significance with F(1, 79) = 12.678, *p* < 0.001, *η*^2^ = 0.138, indicating differences by sex. When evaluating the effects of sex on the test type (pre-/post-test), the results show that males obtained significantly higher pre-test scores than females (F(1, 79) = 11.950, *p* < 0.001, *η*^2^ = 0.131), but not post-test scores (F(1, 79) = 2.926, *p* = 0.091, *η*^2^ = 0.036).

To consider the pre-test differences, a two-way analysis of covariance (ANCOVA) was carried out on the pre-/post-test learning gains, with the sex and conditions as factors and the pre-test as the controlled covariate. It showed a significant single effect of sex on the learning gains with F(1, 76) = 4.309, *p* = 0.041, *η*^2^ = 0.054, meaning that females learned significantly more than males after adjusting for the pre-test scores (see Figure 2). The single effect of the conditions was not significant (F(1, 76) = 0.001, *p* = 0.972), meaning that the learning gains in both conditions (STF and MTF) were similar.

The two-way interaction of sex and the conditions was not significant (F(1, 76) = 0.073, *p* = 0.788, *η*^2^ = 0.001). Pairwise comparisons of sex by condition revealed that females learned more than males in both conditions but this was not significant, with F(1, 76) = 1.694, *p* = 0.197, *η*^2^ = 0.022 for MTF and F(1, 76) = 2.995, *p* = 0.088, *η*^2^ = 0.038 for STF. Also when analyzing the effect of the conditions on sex, non-significant differences were obtained for females (F(1, 76) = 0.027, *p* = 0.870) and males (F(1, 76) = 0.048, *p* = 0.828), meaning that the learning gains were similar across the conditions for males and females.

### 4.2. Sex and Motivation

Table 2 shows the descriptive statistics of the IMI dimensions per sex and condition. The pre- and post-IMI-test values considered measures for the dimensions of interest, perceived competence, effort, no pressure (the reverse of pressure), perceived choice, and value. To assess the internal consistency of the questionnaires, a Cronbach alpha of 0.72 for the pre-IMI-test, 0.69 for the post-IMI-test, and 0.81 for both perception tests were obtained, which can be interpreted in the range of acceptable–good [73].

The differences among the post-IMI and pre-IMI values were analyzed. A three-way repeated-measures ANOVA was carried out to evaluate the effects of the conditions (MTF/STF), sex (male/female), and six IMI dimensions on the pre- and post-IMI delta values.

First, the results show a significant single main effect of sex on the IMI delta values with F(1, 77) = 4.053, *p* = 0.048, *η*^2^ = 0.050, with mean = 0.253 (95% CI [0.113, 0.392]) for males and mean = 0.050 (95% CI [−0.095, 0.194]) for females, meaning that males presented higher gains in the overall IMI scores when considering all six constructs related to motivation as a whole.

The interaction among the IMI dimensions and sex was non-significant (F(5, 73) = 1.648, *p* = 0.158, *η*^2^ = 0.101), the same as the interaction among the IMI dimensions and conditions (F(5, 73) = 0.159, *p* = 0.977, *η*^2^ = 0.011). However, the three-way interaction among the six IMI dimensions, sex, and conditions was significant with F(5, 73) = 2.969, *p* = 0.017, *η*^2^ = 0.169.

The post hoc tests considered the Bonferroni adjustment for multiple comparisons. When analyzing the simple main effect of sex on the IMI dimensions, we found significant differences in competence (F(1, 77) = 7.195, *p* = 0.009, *η*^2^ = 0.085) and choice (F(1, 77) = 4.407, *p* = 0.039, *η*^2^ = 0.054) favoring males over females, as Figure 3 shows. For the rest of the IMI dimensions, no significant differences were obtained.

Now, when analyzing the three-way interaction among sex, the IMI dimensions, and the conditions, we determined that, under the MTF condition (see Figure 4), there were no significant differences in sex for the IMI competence construct (F(1, 77) = 0.002, *p* = 0.963), while, for the choice dimension, males presented a significantly higher gain than females (F(1, 77) = 5.982, *p* = 0.017, *η*^2^ = 0.072). On the contrary, under the STF condition (see Figure 5), males obtained significantly higher gains for competence (F(1, 77) = 13.956, *p* < 0.001, *η*^2^ = 0.153), while no relevant differences existed between the males and females for perceived choice (F(1, 77) = 0.280, *p* = 0.598).

## 5. Discussion

### 5.1. RQ1: What Are the Differences in the Learning Outcomes between Male and Female Students Using MatematicaST?

When analyzing RQ1, the results by sex are not similar in terms of the learning gains. Females presented higher learning gains than males when adjusting for the pre-test scores. At first, our findings seemed to support past research suggesting that females present more engagement and learning gains than males under DGBL [4,8,20], and they seemed to contradict research showing no significant sex differences in learning [10,11]. However, it is relevant to consider that the males had higher pre-test scores than the females, and that the post-test scores were similar between the sexes. Hence, we can consider that the females’ outperforming learning gains were at least partially supported by their lower prior knowledge level, as diverse studies state the idea that lower levels of prior knowledge provide more space for learning improvements [22,34,38,74].

Now, from the perspective of the instructional feedback conditions provided in the present experiment, the multiple-try and single-try game versions provided similar results in terms of the learning gains for each sex. Therefore, the reported sex effects on learning seem to have not been affected by these instructional feedback conditions. Furthermore, the present results show no learning differences among the two instructional conditions involved in this experiment, namely, STF and MTF. Such results contradict past research supporting the idea that MTF outperforms STF in high-order or complex learning outcomes that are not just verbatim tasks [33,44,57,68].

A factor that could have affected the results is the presence of an overall high pre-test score (and, hence, a high prior knowledge level) in general when contrasted to other literature. While past literature usually presents pre-test levels in the range of 50–80% [42,45,46,47], in our case, the average score in the pre-test was 84%, and in males under the MTF game version, it was 95%, possibly affecting the possibility of learning improvements for this group.

### 5.2. RQ2: What Differences Exist in Motivation in Terms of Interest (RQ2a), Competence (RQ2b), and Autonomy (RQ2c) between Male and Female Students Using MatematicaST?

When looking at sex perception in terms of interest (RQ2a), no differences were assessed between the sexes. The results support Manero et al. [24], who found no interest differences by sex, and they partially contradict Chung and Chang’s study [11], which states that “in a moderate genre digital game, female learners’ motivation is significantly higher than that of male learners”. This is because, from the perspective of learning, our game does appear to have had a moderate sex influence, while, in terms of motivation, it does not. This could be due to the basic mechanics of the game, as it does not require much coordination skill (as in a shooting- or jumping-platform game). This may have benefited the females, as evidenced in [3,11], as they prefer exploration mechanics over competitive ones. Furthermore, the gamification elements used in our MatematicaST game were points, rankings, and levels, which foster progression and competition dynamics. These elements and their associated mechanics could have negatively affected the females due to the absence of other game elements, such as the challenges, storylines, and fun interaction found in Chung and Chang’s game [11]. However, despite the general belief that males find games more relaxing and engaging than females [16], our research revealed that the males did not perceive MatematicaST as more interesting. This discrepancy may be attributed to boys’ preferences for action games, as highlighted by Khan et al. [8], whereas MatematicaST employs a point-and-click mechanic.

Regarding the competence (RQ2b) and autonomy (choice) (RQ2c) dimensions, both sexes reported positive increments. However, the males showed significantly higher increments compared to the females in both constructs. Despite the females’ higher learning gains, the males presented higher perceived competence and autonomy, both with medium effect sizes. Such results do not follow the general idea proposed by SDT-CET that competence and autonomy are strong predictors of motivation and learning [59,60,61,62]. These contradicting results may be attributed to the Dunning–Kruger effect [75], where students tend to significantly overestimate their performance, fostering a belief in their adequate knowledge of a given topic.

Now, from the perspective of the females, despite having higher learning gains, they perceived themselves as less competent compared to the males. Such results are coherent with Goetz et al. [31] regarding “students’ beliefs about their competence in mathematics, with female students reporting lower perceived competence than male despite having the same average grades.” From a social learning theory perspective [76], such sex differences in math competence perceptions reveal that there is still a strong cultural bias that identifies math as a predominantly male domain due to stereotypes. Past research has recognized that gender math stereotypes negatively affect women’s mathematics achievements, primarily through their self-concept [77], leading them to underestimate their mathematical abilities and experience more mathematics anxiety than boys [78]. Furthermore, although boys continue to outperform girls in mathematics in many countries, these gaps have narrowed or even shifted in favor of female students in the last two decades [79].

As competence and autonomy are strong predictors of intrinsic motivation, we can infer that the males felt significantly more motivated by the activity. This is corroborated by the overall higher IMI scores of the males compared with the females when considering all six motivational constructs together, and with almost a medium effect size. Perhaps this motivation was due to the gaming itself and that males tend to be more competitive than females at that age, especially in computer gameplay scenarios. Also, the points mechanism and the leaderboard could have positively affected the males’ motivation. Our findings support Bonanno and Kommers [16], who reported that boys consider playing games a memorable and unique experience. Also, males’ high pre-test and post-test scores could make them feel confident in their knowledge of the math topics involved, increasing their perception of their competence and autonomy while reducing their interest in a subject that they feel they have already mastered.

### 5.3. RQ3: What Are the Differences in Perceptions of Effort (RQ3a), No Pressure (RQ3b), and Value (RQ3c) between Male and Female Students Using MatematicaST?

Concerning the effort construct (RQ3a), no significant differences were observed between the males and females. Also, both sexes showed decreasing pre-/post-test delta values, which indicate that the students ended up needing to work less than they were willing to at the beginning of the activity. Although the pre-/post-test difference is negative, the post-test level remains high (80%), providing indications of the non-existence of factors perceived as controlling by the students [59], which could have negatively affected their motivation and learning. An interesting situation is that the females showed a decrease in effort despite perceiving themselves as less competent when compared to the males and outperformed them in learning gains. An interpretation of this phenomenon might be rooted in cultural factors. It is plausible that the females perceived their mathematical abilities to be lower than they actually were, causing them to make more effort than was required in the mathematical tasks. This misperception has been documented in previous studies related to sex stereotypes in mathematics [77,79].

Regarding the no-pressure construct (RQ3b), no significant differences existed between the males and females. By considering pressure and anxiety as related constructs, the results support findings from past gender math anxiety studies [31], indicating that no sex differences exist for state anxiety (that is, anxiety during specific activities, such as, in this case, participating in a math training activity with a digital game). Both sexes obtained decreases in their efforts and increases in the no-pressure dimensions, corroborating the fact that both intervention conditions were suitable to the promotion of learning [59].

In terms of the value dimension (RQ3c), although there were no significant differences between the sexes, the males presented higher levels than the females. This is aligned with the idea that females do not see games as a unique experience but rather as another alternative for learning, and they find them less useful compared to males [16].

### 5.4. RQ4: What Differences Exist in the Effects of Sex between the Two Feedback Conditions (MTF and STF)?

As indicated earlier, there were no learning differences between the two instructional conditions, MTF and STF, both overall and when evaluated for each sex. However, upon examining the potential effects on the motivational variables resulting from the interaction of sex and the instructional conditions, the results revealed significant differences. Specifically, it was found that in the MTF condition, there were no significant sex differences regarding the sense of competence, but there were differences in terms of autonomy. Conversely, in the STF condition, sex differences were present for competence but not for the autonomy construct.

These results are relevant, as they suggest a possible sex difference between the two instructional implementations (MTF and STF) used in our experiment. They directly indicate that when using multiple attempts, the females felt as competent as the males in the exercise activity, but, at the same time, they felt less free or autonomous than the males. One possible explanation could stem from the fact that females perceive themselves as less competent in math despite achieving similar learning outcomes [31]. Therefore, the mechanic of having multiple attempts (MTF) to answer the same exercise makes the student face an item they know they have answered incorrectly, generating a degree of frustration [49,54,55] and reinforcing the idea of limited effectiveness for an item that needs repetition. If we add to this the fact that the pre-test level (and therefore, prior knowledge) was lower in the females, we can assume that they probably used the retry mechanism more often than the males when not answering correctly on the first attempt. This aligns with the idea that low achievers tend to feel more frustrated when using MTF [55,56,57].

This study contributes valuable insights into sex differences in DGBL, emphasizing the need for gender-responsive curriculum designs and evidence-based policies. Curriculum developers should strike a balance between autonomy and competence. For girls, this means fostering confidence while maintaining high standards of knowledge acquisition.

Our findings suggest that girls may feel less confident or empowered when given multiple attempts to answer questions. Thus, curriculum developers could consider strategies to boost girls’ autonomy, such as providing clear guidelines, encouraging self-assessment, and fostering a growth mindset. The study findings also suggest that girls might struggle more when they have only one chance to answer questions. Therefore, curriculum designers could explore ways to enhance girls’ competence, such as by providing adaptive feedback based on sex-specific needs (for example, implementing additional elaborated feedback (e.g., hints, worked-out examples) for girls during the single attempt).

Policymakers should recognize sex differences in DGBL and promote equity. Policies can encourage schools to adopt inclusive practices that cater to diverse learning needs. For instance, allocating resources for sex-sensitive game-based learning initiatives can foster equal opportunities. Also, policies should emphasize teacher training in sex-responsive pedagogy. Educators need awareness and skills to adapt their teaching methods to accommodate sex-specific preferences.

Regarding the study limitations, as age is an important personal trait in educational contexts [34,38], it could have affected the multiple attempts’ effectiveness and the sex differences found in this work. Therefore, future studies should include subjects from other levels, from preschoolers [46] and secondary-school students to tertiary-education students. Also, the elderly outside formal learning should be considered, with games aiding their cognitive and emotional needs [80]. Other limitations arise from the nature of the learning objectives used. Although they were of high complexity, future research should include LOs from other mathematical domains. Differences in students’ socioeconomic statuses (SESs) can be considered another possible limitation, as the study involved private and subsidized schools but not public ones.

Likewise, these results open up the possibility of questioning whether there might be other instructional elements and combinations thereof that also generate sex differences, either in terms of learning or motivation. From this, it becomes necessary to move forward with further analyses. These studies should initially focus on replicating these results and then proceed to evaluate the various factors and interactions that generate different effects in both sexes when varying the number of attempts.

## 6. Conclusions

The present study provided empirical results on the different effects obtained by sex when using MatematicaST, a digital math game for primary students. The results indicate that the females outperformed the males in learning gains, despite presenting lower motivation levels, especially regarding the self-perceptions of competence and autonomy. In this way, the present research contributes to the literature not only by providing a successful case of game-based learning favoring females in terms of learning, but also by giving insights into the internal emotional processes that could affect such differences in learning.

This study is perhaps the first research providing empirical evidence on the effects of multiple attempts by sex. An interaction effect between sex and the feedback condition was found. The females under multiple-try feedback presented lower autonomy levels than the males, while for STF, there were no differences. On the contrary, the females under MTF presented similar competence levels to those of the males, but under STF, the females presented lower levels. Such findings suggest that not all feedback types are sex-neutral, especially the ones involving multiple attempts. It will be relevant to check whether these findings extend to other learning contexts, and to delve deeper into the factors that enable the generation of such sex differences.

The practical implications regard the use of multiple attempts not only in automated assessments and testing but also in game-based formative activities. The use of multiple attempts in existing LMS (e.g., Moodle) activities will generate learning gains in students, especially in females. Also, the design of future educational computer systems should include multiple attempts, but while considering the possible sex differences leveraged in this study. By acknowledging these findings, educators and policymakers can create more effective and equitable learning environments for all students.

Future research directions should also include multiple-try implementations with various attempt numbers, instructional feedback types, and game mechanics. Also, possible sex differences in multiple-try use and its outcomes under all these variants need further study. Finally, further studies that incorporate other demographic variables that may influence the effectiveness of DGBL, such as age, income level, race, location, parent’s employment, and level of education, are needed. 

## Figures and Tables

**Figure 1 behavsci-14-00488-f001:**
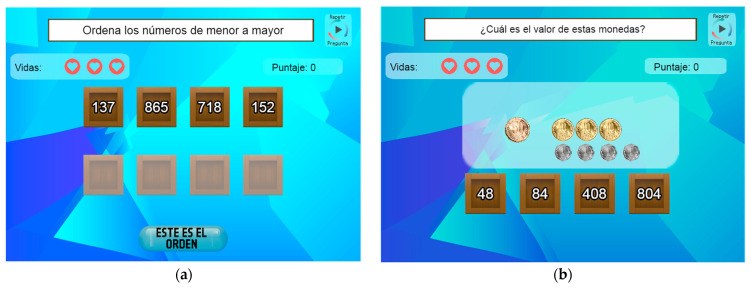
(**a**) Number ordering mini-game; (**b**) money counting mini-game.

**Figure 2 behavsci-14-00488-f002:**
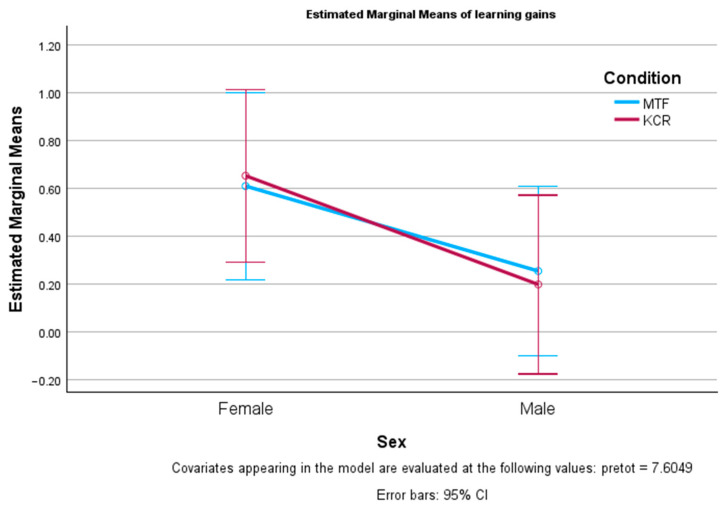
Pre-/post-test learning differences per sex and feedback condition after controlling for the pre-test.

**Figure 3 behavsci-14-00488-f003:**
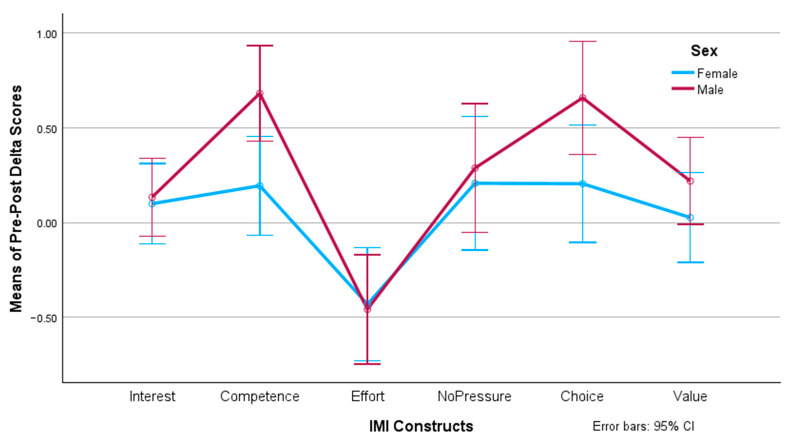
Pre-/post-test delta values of motivational constructs per sex.

**Figure 4 behavsci-14-00488-f004:**
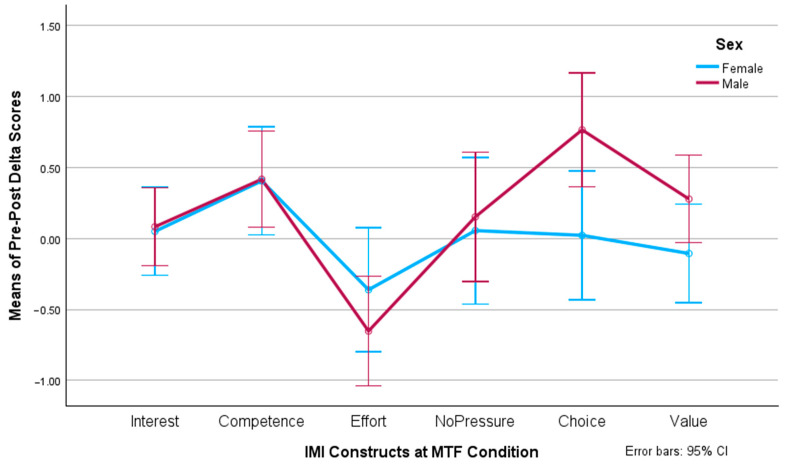
Pre-/post-test delta values of motivational constructs per sex for the MTF condition.

**Figure 5 behavsci-14-00488-f005:**
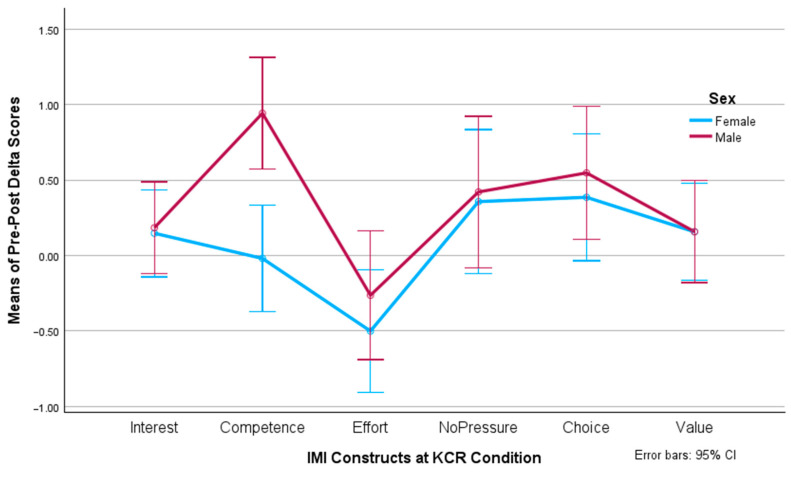
Pre-/post-test delta values of motivational constructs per sex for the STF condition.

**Table 1 behavsci-14-00488-t001:** Pre-test, post-test, and learning-gain descriptive statistics per sex and condition.

Sex	Condition	Pre-Test		Post-Test		Gain		N
		Mean	SD	Mean	SD	Mean	SD	
Female	MTF	6.89	2.16	7.67	1.99	0.78 ***	0.94	18
	STF	6.99	1.97	7.79	1.55	0.80 ***	1.00	21
	Total	6.94	2.03	7.73	1.74	0.79 ***	0.96	39
Male	MTF	8.54	0.80	8.58	0.83	0.03	0.74	23
	STF	7.83	1.53	7.97	1.57	0.14	0.94	19
	Total	8.22	1.22	8.30	1.25	0.08	0.82	42
Total	MTF	7.82	1.74	8.18	1.51	0.36 **	0.90	41
	STF	7.39	1.80	7.88	1.54	0.49 **	1.01	40
	Total	7.60	1.77	8.03	1.52	0.42 ***	0.95	81

MTF: multiple-try feedback. STF: single-try feedback. ** *p* < 0.01, *** *p* < 0.001.

**Table 2 behavsci-14-00488-t002:** IMI perception constructs’ descriptive statistics per condition.

IMI Construct	Sex	Condition	Pre-Test		Post-Test		Gain		N
			Mean	SD	Mean	SD	Mean	SD	
Interest	Female	MTF	4.64	0.48	4.69	0.73	0.05	0.75	18
		STF	4.70	0.56	4.85	0.38	0.15	0.69	21
		Total	4.67	0.52	4.77	0.56	0.10	0.71	39
	Male	MTF	4.56	0.60	4.64	0.67	0.08	0.63	23
		STF	4.58	0.59	4.77	0.44	0.18	0.59	19
		Total	4.57	0.58	4.70	0.57	0.13	0.60	42
	Total	MTF	4.59	0.54	4.66	0.69	0.07	0.68	41
		STF	4.65	0.57	4.81	0.40	0.17	0.63	40
		Total	4.62	0.55	4.74	0.57	0.12	0.65	81
Competence	Female	MTF	3.96	0.83	4.38	0.74	0.41 *	0.82	18
		STF	4.29	0.75	4.29	0.92	−0.02	0.72	21
		Total	4.14	0.80	4.33	0.83	0.18	0.79	39
	Male	MTF	4.13	0.78	4.56	0.69	0.42 *	0.82	23
		STF	3.83	0.86	4.75	0.46	0.94 ***	0.90	19
		Total	4.00	0.82	4.65	0.60	0.65 ***	0.88	42
	Total	MTF	4.06	0.80	4.48	0.71	0.41 **	0.81	41
		STF	4.07	0.83	4.51	0.77	0.44 ***	0.93	40
		Total	4.06	0.81	4.49	0.73	0.42 ***	0.87	81
Effort	Female	MTF	4.56	0.75	4.19	0.89	−0.36	0.74	18
		STF	4.52	0.84	4.02	0.99	−0.50 *	0.87	21
		Total	4.54	0.79	4.10	0.94	−0.44 **	0.80	39
	Male	MTF	4.50	0.78	3.85	0.98	−0.65 **	0.99	23
		STF	4.55	0.71	4.29	0.79	−0.26	1.07	19
		Total	4.52	0.74	4.05	0.92	−0.48 **	1.04	42
	Total	MTF	4.52	0.76	4.00	0.95	−0.52 ***	0.89	41
		STF	4.54	0.77	4.15	0.90	−0.39 *	0.96	40
		Total	4.53	0.76	4.07	0.92	−0.46 ***	0.93	81
No Pressure	Female	MTF	3.92	0.79	3.97	0.90	0.06	1.26	18
		STF	4.05	1.00	4.40	0.90	0.36	0.91	21
		Total	3.99	0.90	4.21	0.92	0.22	1.08	39
	Male	MTF	3.67	0.91	3.83	1.24	0.15	1.21	23
		STF	3.71	0.73	4.13	1.10	0.42 ^†^	0.98	19
		Total	3.69	0.83	3.96	1.18	0.27 ^†^	1.11	42
	Total	MTF	3.78	0.86	3.89	1.09	0.11	1.22	41
		STF	3.89	0.89	4.27	1.00	0.39 *	0.93	40
		Total	3.83	0.87	4.08	1.06	0.25 *	1.09	81
Choice	Female	MTF	3.96	0.90	3.99	0.66	0.02	0.92	18
		STF	3.70	1.10	4.10	0.75	0.39 ^†^	0.97	21
		Total	3.82	1.01	4.05	0.70	0.22	0.95	39
	Male	MTF	3.34	1.16	4.12	0.75	0.77 ***	0.95	23
		STF	3.44	0.89	3.99	0.95	0.55 *	1.03	19
		Total	3.38	1.04	4.06	0.84	0.67 ***	0.98	42
	Total	MTF	3.61	1.09	4.07	0.70	0.44 **	1.00	41
		STF	3.57	1.00	4.05	0.84	0.46 **	0.99	40
		Total	3.59	1.04	4.06	0.77	0.45 ***	0.98	81
Value	Female	MTF	4.52	0.62	4.42	0.88	−0.11	0.69	18
		STF	4.63	0.60	4.79	0.37	0.16	0.65	21
		Total	4.58	0.60	4.62	0.67	0.04	0.67	39
	Male	MTF	4.46	0.52	4.74	0.47	0.28 ^†^	0.53	23
		STF	4.34	0.50	4.50	0.94	0.16	1.04	19
		Total	4.41	0.51	4.63	0.72	0.22 ^†^	0.80	42
	Total	MTF	4.49	0.56	4.60	0.69	0.11	0.63	41
		STF	4.49	0.57	4.65	0.71	0.16	0.85	40
		Total	4.49	0.56	4.62	0.70	0.13	0.74	81

MTF: multiple-try feedback. STF: single-try feedback. ^†^ *p* < 0.10, * *p* < 0.05, ** *p* < 0.01, *** *p* < 0.001.

## Data Availability

The data presented in this study are available upon request from the corresponding author. The data are not publicly available as they may present identification data of the study subjects.

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
