# Peer review of "A Digital Math Game and Multiple-Try Use with Primary Students: A Sex Analysis on Motivation and Learning"

_behavsci, 2024, doi:10.3390/bs14060488_

Round 1

Reviewer 1 Report

Comments and Suggestions for Authors

Please have a look at the attached pdf file.

Author Response

All corrections have been applied. 

See the attached file for details

Reviewer 2 Report

Comments and Suggestions for Authors

This paper provides an insightful exploration into the impacts of gender on motivation and performance using a math digital game in primary school students. The study is well-grounded in the literature on gender differences and digital game-based learning, and it presents a significant amount of empirical data to support its findings. The methodology is robust, with clear descriptions of the instruments and statistical analyses used. Nevertheless, there are areas that could be enhanced to strengthen the paper:

  1. Introduction and Literature Review (Sections 1-2):

    • The introduction effectively sets up the problem and the need for the study. Furthermore, it could benefit from a more concise overview of existing research gaps. It would be helpful to specifically state early on how this study aims to fill these gaps.
    • In the literature review, while the coverage of gender differences in technology use and learning is comprehensive, the transitions between studies are somewhat abrupt. Therefore, it would enhance readability to link these studies more smoothly, showing how they build on one another to justify this study’s focus.
    • Methodology (Section 3):

      • The selection of the sample and the design of the study are well justified and detailed. Nevertheless, the explanation of why these particular schools and grades were chosen could be elaborated upon to strengthen the external validity of the study.
      • It is recommended to clarify any potential biases or limitations introduced by the study design or sample selection at this stage, rather than only addressing them in the discussion.
    • Results (Section 4):

      • The results are clearly presented with appropriate statistical analyses. Including visual aids such as graphs or charts to depict the key differences in performance and motivation by gender across different conditions would enhance understanding and impact.
      • The discussion on the implications of the findings is insightful. However, it might be beneficial to discuss alternative explanations for gender differences observed and how they might interact with the cultural or educational context of the participants.
    • Discussion and Conclusion (Sections 5-6):

      • The discussion section does a good job of linking back to the literature and discussing the study’s findings in the context of existing research. It would be strengthened by a more critical examination of how these findings might be applied in practical settings, such as in curriculum development or educational policy.
      • The conclusion succinctly summarizes the findings but could be expanded to suggest future research directions more specifically, particularly in terms of investigating other demographic variables that may influence the effectiveness of DGBL.

Overall Evaluation: This paper makes a valuable contribution to the field of educational technology and gender studies. The findings are significant and warrant further discussion and exploration in broader educational contexts. With the suggested revisions, this paper could provide a strong foundation for future research and practical application in enhancing gender-inclusive educational practices.

Comments on the Quality of English Language

Found some typographical and grammatical error. 

Author Response

All of the mentioned corrections have been considered and incorporated into the manuscript.  See the attached file for details.

Reviewer 3 Report

Comments and Suggestions for Authors

See attached file for comments, suggestions, and ideas.

Comments on the Quality of English Language

See comments in the attached file

Author Response

corrections and suggestions applied to the manuscript.

see the attached table for details.

Reviewer 4 Report

Comments and Suggestions for Authors

1.        Although the article encompasses a wide range of studies, there is room for improvement in the manner in which these findings are incorporated into a unified narrative that directly addresses the study's hypothesis. At times, there appears to be a precipitous transition between various research findings and their pertinence to the objectives of the present study.

2.     Kindly elaborate on the treatment. Including additional information regarding the administration of the treatment facilitates the replication of the study by other researchers. It should contain precise information regarding the location, duration, and characteristics of the interventions.

3.     The paper exhibits commendable consistency in its terminology choices, particularly with regard to educational technology and gender-related aspects. In order to bolster the paper's rigour, it is recommended that the usage of terms such as 'gender' and'sex' be maintained consistently with the most recent academic standards.

4.     Minimal typographical and grammatical errors are present, suggesting that the document has been meticulously proofread.

5.     Nevertheless, the paper would be enhanced by a comprehensive line-by-line revision in order to refine sentence structures and guarantee impeccable adherence to scholarly writing norms.

6.     While the discussion connects findings with previous studies, it could benefit from a deeper exploration of why these gender differences might exist beyond the literature cited. For example, considering psychological theories of gender development or educational biases could add depth.

7.     To enhance its comprehensiveness, the conclusion might contemplate the ramifications of the research that extend beyond its immediate educational milieu. This may encompass recommendations for modifications to policies, ramifications for creators of digital content, or more extensive effects on society.

Author Response

All the suggestions have been incorporated to the manuscript

see the attached file for details.

Round 2

Reviewer 4 Report

Comments and Suggestions for Authors

The authors have revised the manuscript thoroughly and successfully improved the quality of the paper. Congrats